# Characterisations of adverse events detected in a university hospital: a 4-year study using the Global Trigger Tool method

Hans Rutberg,[1] Madeleine Borgstedt Risberg,[2] Rune Sjödahl,[3,4] Pernilla Nordqvist,[4] Lars Valter,[2] Lena Nilsson[5]

For numbered affiliations see end of article.

**Correspondence to**
Dr Lena Nilsson;
lena.nilsson@lio.se

## ABSTRACT

**Objectives:** To describe the level, preventability and categories of adverse events (AEs) identified by medical record review using the Global Trigger Tool (GTT). To estimate when the AE occurred in the course of the hospital stay and to compare voluntary AE reporting with medical record reviewing.

**Design:** Two-stage retrospective record review.

**Setting:** 650-bed university hospital.

**Participants:** 20 randomly selected medical records were reviewed every month from 2009 to 2012.

**Primary and secondary outcome measures:** AE/1000 patient-days. Proportion of AEs found by GTT found also in the voluntary reporting system. AE categorisation. Description of when during hospital stay AEs occur.

**Results:** A total of 271 AEs were detected in the 960 medical records reviewed, corresponding to 33.2 AEs/1000 patient-days or 20.5% of the patients. Of the AEs, 6.3% were reported in the voluntary AE reporting system. Hospital-acquired infections were the most common AE category. The AEs occurred and were detected during the hospital stay in 65.5% of cases; the rest occurred or were detected within 30 days before or after the hospital stay. The AE usually occurred early during the hospital stay, and the hospital stay was 5 days longer on average for patients with an AE.

**Conclusions:** Record reviewing identified AEs to a much larger extent than voluntary AE reporting. Healthcare organisations should consider using a portfolio of tools to gain a comprehensive picture of AEs. Substantial costs could be saved if AEs were prevented.

### Strengths and limitations of this study

- The sample is representative of the care given at the university hospital.
- The review team was experienced and remained the same throughout the study.
- The study was conducted in a single hospital which may restrict the generalisability of the findings.

Institute for Healthcare Improvement (IHI), is widely used for retrospective reviews of medical records.[6 7] The GTT can be used as a quality improvement tool in clinical practice to estimate and track AE rates over time. Its aim is to enable longitudinal comparisons and assessment of implemented patient safety measures and support the identification of target areas for improvement.[6] The Swedish version of the GTT was published in 2008 and includes evaluation of preventability of harm. The same preventability assessment was used in a study of the incidence of AEs in Swedish hospitals.[8] The Swedish handbook includes a list of different categories of harm (hospital-acquired infections, falls, pressure ulcers, etc).

At the University Hospital in Linköping, in southeast Sweden, the GTT method has been applied since 2009 with a monthly review of 20 randomly selected medical records. The hospital is a middle-sized university hospital with about 650 beds and 32 000 admissions yearly. It has most medical specialties including neurosurgery and cardiac surgery but no transplantation service.

In Sweden, it is mandatory to report severe AEs to the National Board of Health and Welfare, but all hospitals also have a local system for voluntary AE reporting by the providers. AEs, incidents and near misses are reported.

## INTRODUCTION

Several methods have been used to identify and measure medical adverse events (AEs), including voluntary reports, mining of administrative databases, patient claims and medical record reviews.[1–5] The Global Trigger Tool (GTT), developed by the

The aim of this study was to describe the level, preventability and characteristics of AEs in a Swedish University Hospital, the latter by using a national harm classification list. We hypothesised that patients with an AE would have a prolonged hospital stay and by thorough examination of the cases where harm was identified we tried to estimate when the AEs occurred in the course of the hospital stay. Furthermore, we wanted to compare voluntary AE reporting with a medical record review for AE detection.

## METHODS
### Setting

Twenty randomly selected medical records from all departments of the University Hospital in Linköping, except the paediatric and psychiatric departments and the obstetric ward, were reviewed every month for a 4-year period from 2009 to 2012. The randomly selected hospital stay that was reviewed is referred to as the index admission. All departments included in this study use the same electronic medical record system.

According to the policy activities that constitute research at County Council of Östergötland, this work met the criteria for operational improvement activities exempt from ethics review.

### Review process

The GTT review followed the IHI methodology, that is, a two-stage review, with two nurses as the primary stage reviewers and one of two physicians as the secondary stage reviewer.[6] In the first stage, a time limit of 20 min/record was applied.

We used the IHI GTT definition for harm: unintended physical injury resulting from or contributed to by medical care that requires additional monitoring, treatment or hospitalisation, or that resulted in death.[6]

The physicians made the final decision together with the nurses on the presence or absence of an AE, its severity and potential preventability. The reviews during the 4-year period were carried out by a team consisting of three experienced registered nurses and two experienced physicians, all with expertise in the field of patient safety. One of the physicians was a senior anaesthesiologist and the other a senior surgeon.

Patient harm severity was categorised according to the National Coordinating Council for Medication Error Reporting and Prevention index (NCC MERP) on a scale from E to I, where E is a temporary harm that requires intervention, F a temporary harm that requires initial or prolonged hospitalisation, G a permanent harm, H a harm that requires intervention to sustain life and I a harm that contributes to the patient's death.[6] The injury identified was also classified into different harm categories according to the classification list in the Swedish handbook (table 1). The patient records were also categorised regarding predominantly surgical (all operating specialties) or medical care. According to the

| Table 1 | Harm classification |
| --- | --- |
| **Class** | **Harm** |
| Care | |
| 1 | Allergic reaction |
| 2 | Bleeding, not in connection with surgery |
| 3 | Fall |
| 4 | Thrombosis |
| 5 | Pressure ulcers (grades 2–4) |
| 6 | Distended urinary bladder |
| 7 | Thrombophlebitis |
| Hospital-acquired infections | |
| 8 | Central venous catheter infection |
| 9 | Pneumonia (not ventilator-associated pneumonia) |
| 10 | Postoperative wound infection |
| 11 | Sepsis |
| 12 | Urinary tract infection |
| 13 | Ventilator-associated pneumonia |
| 14 | Other hospital-acquired infection |
| Surgical injury | |
| 15 | Wrong site surgery |
| 16 | Injury of organ during operative procedure |
| 17 | Postoperative bleeding/haematoma (not requiring reoperation) |
| 18 | Reoperation |
| 19 | Other surgical complication |
| Others | |
| 20 | Cardiac or pulmonary failure or arrest |
| 21 | Anaesthesia-related injury |
| 22 | Medication-related injury |
| 23 | Medical device-related injury |
| 24 | Obstetric injury |
| 25 | Neurological injury |
| 26 | Other injury |

GTT method, the total length of stay in hospital for patients with or without an AE was calculated, with the day of admission and the day of discharge counted as two separate days. We also made an additional review of the records whereby we identified harm to evaluate when the AE occurred in the course of the hospital stay.

Preventability was graded on a scale from 1 to 6 where 1 indicates virtually no evidence for preventability and 6 indicates virtually certain evidence for preventability. At a rating of at least 4 (ie, more than 50% likelihood), AEs were classified as preventable.[8]

The total number of hospital admissions with the same inclusion criteria as the random sample for GTT review was calculated for the 4-year period.

The voluntary AE reporting system (Synergi Life, DNV GL, Høvik, Norway) was introduced at the university hospital in 2004 and approximately 7000 AEs are reported annually. It is a web-based IT system where all employees have access to report suggestions for improvements, identified risks and AEs. Hospital-acquired infections are also reported in the system. The staff can also take note of all reports from their own department. Patients and relatives can also report in the system by a separate open access. The Synergi Life system is used in several

Swedish counties. Whenever harm was identified, the voluntary reporting system was checked to see if the AE was included. This could be achieved by searching in the Synergi Life system for reports on the date for the AE and/or the patients' birth data. The reporting system includes a brief description of the AE and a heading, and often also patient data.

## Statistical analysis

Descriptive statistics included frequencies (%), means and SDs. The $\chi^2$ test was used to determine if there were any statistical differences between the various groups. For all analyses, $p < 0.05$ was regarded as statistically significant.

Statistical software IBM SPSS V.20 was used for all statistical analyses and data processing.

## RESULTS

A total of 960 medical records were reviewed using the GTT method (473 women and 487 men). The mean age for women was 66.1 years (range 18–95 years) and for men 65.9 years (range 18–96 years). The distribution of medical records reviewed for the different age groups is shown in table 2. Forty-eight per cent of all records reviewed were categorised as surgical and 52% as medical care.

A total of 271 AEs were detected in 197 patients among the 960 medical records reviewed. The number of AEs/1000 patient-days was 33.2. Overall, 20.5% of all patients reviewed experienced at least one AE. Seventy-one per cent of the AEs were assessed as preventable. Six per cent of AEs were classified according to preventability grade 1, virtually no evidence for preventability; 9% in grade 2, slight-to-modest evidence for preventability; 14% in grade 3, preventability not likely, less than 50–50 but close; 56% in grade 4, preventability more likely than not, more than 50–50 but close call; 14% in grade 5, strong evidence for preventability and 1% in grade 6, virtually certain evidence of preventability.

The proportion of AEs in the different age groups is shown in figure 1. During the 4-year period, no significant change in the rate of AEs was seen within or between the groups. No statistical difference in AE rate was seen between men and women or between patients older than 65 years and younger patients.

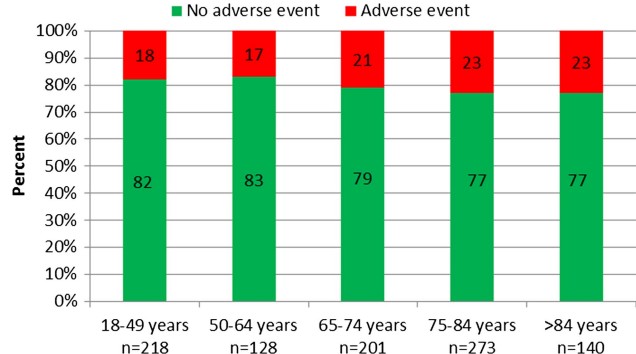

**Figure 1** The proportion of patients with or without an adverse event during the years 2009–2012.

In our study, 10 patients (5.1%) had an AE that was identified on index admission but had occurred within 30 days before admission. An additional 18 patients (9.1%) were admitted due to an AE that was caused by healthcare received or initiated more than 30 days before the index admission; for example, bleeding caused by warfarin. In 40 patients (20.3%), the harm occurred during the index hospital stay but was detected within 30 days after discharge either in primary care or in connection with readmission. The remaining 129 patients (65.5%) had an AE that occurred and was detected during the index hospital stay. Table 3 shows the distribution of AEs according to severity.

AEs were more common in surgical care than in medical care (p<0.001). Twenty-six per cent of patients undergoing surgical care had at least one AE. Fifteen per cent of patients receiving medical care had at least one AE. The distribution of the 271 AEs in the different harm categories according to the Swedish handbook for medical record reviewing divided in surgical and medical care is shown in figure 2. Of the in total 120 hospital-acquired infections identified, the most common types were postoperative wound infections (40%), urinary tract infections (21%), ventilator-associated pneumonia (6%) and central venous catheter infections (5%).

Patients who experienced an AE had a longer hospital stay. The total length of stay for patients without an AE was 7.4 (SD=12.5 days) and 12.8 days (SD=12.9 days) for patients with an AE. Total length of stay for patients with and without an AE for the different age groups is shown in figure 3.

Detailed examination of the 129 medical records where an AE occurred and was detected during the

| Table 2 | Number of medical records in the different age groups reviewed with the GTT method during 2009–2012 |
|---------|---------|---------|---------|---------|---------|---------|
| | **Number of records in each age group** | | | | | |
| | **18–49 years** | **50–64 years** | **65–74 years** | **75–84 years** | **>84 years** | **Total** |
| Women | 112 | 56 | 100 | 125 | 80 | 473 |
| Men | 106 | 72 | 101 | 148 | 60 | 487 |
| Total | 218 | 128 | 201 | 273 | 140 | 960 |
| GTT, Global Trigger Tool. | | | | | | |

**Table 3** The distribution of AEs according to the NCC MERP severity scale

| Harm score | Description | AEs Frequency | Per cent |
|---|---|---|---|
| E | Patient experienced temporary harm that required intervention | 118 | 43.5 |
| F | Patient experienced temporary harm that required initial or prolonged hospitalisation | 134 | 49.4 |
| G | Patient experienced permanent harm | 5 | 1.8 |
| H | Patient experienced harm that required life-sustaining intervention | 6 | 2.2 |
| I | Patient died as a result of the harm | 8 | 3.0 |
| Total | | 271 | 100 |

AE, adverse event; NCC MERP, National Coordinating Council for Medication Error Reporting and Prevention.

hospital stay revealed that an AE usually occurred in the early period of the hospital stay; 61.2% of the AEs detected occurred on day 1–4, 24% occurred on day 5–8, 8.5% on day 9–12 and the remaining 6.2% occurred on day 13 or later.

During the 4-year period (2009–2012), there were 128 100 admissions to the hospital with the same inclusion criteria as the GTT sample. During the same period, 24 834 AEs, incidents and near misses were reported in the voluntary reporting system. Of the 271 AEs identified with the GTT method only, 17 (6.3%) were reported in the voluntary AE reporting system.

## DISCUSSION

In summary, our study shows that only 6.3% of the AEs detected by the GTT were reported by the staff. Hospital-acquired infections were the most common AE category. The AEs occurred and were detected during the hospital stay in 2/3 of cases; the rest occurred or were detected within 30 days before or after the hospital stay.

One of the main findings of this study was that only a few of the AEs identified by the review of the medical records were reported voluntarily. This is in accordance with what has been reported by others. In a study from the Mayo Clinic, it was reported that 27.7% of discharges reviewed with GTT were discovered to have experienced an AE. When provider-reported events were used for identification of AEs, only 5% of patients were found to have an AE.[1] Classen et al[2] reported that GTT found at least 10 times as many AEs as voluntary reporting and the Agency for Healthcare Research and Quality patient safety indicators.

Voluntary reporting is one of the cornerstones of patient safety practice and is commonly used in most hospitals. However, even though reporting systems are considered fundamental for improving safety in health care, several studies have also documented under-reporting of serious AEs.[1–5] The strength of adverse reporting systems is identification of risks and near misses, and the findings in this and other studies

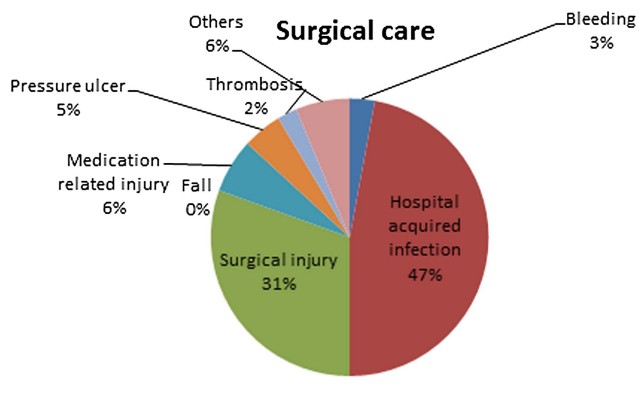

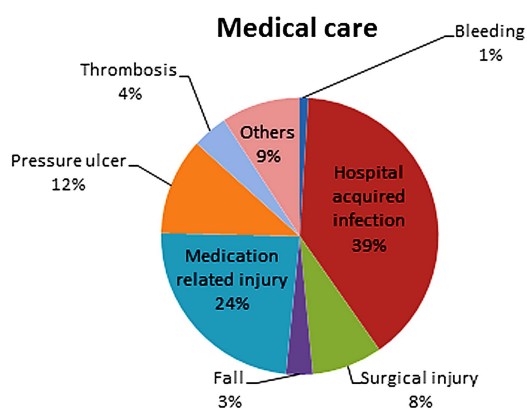

**Figure 2** The distribution of adverse events during the years 2009–2012 in surgical care (n=174) and medical care (n=97) in different harm categories.

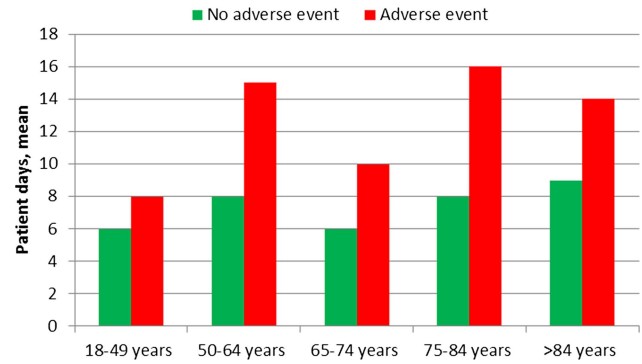

**Figure 3** Length of hospital stay for 197 patients with adverse event (AE) and 763 patients with no AE during 2009–2012.

indicate that reporting systems should be supplemented by the detection of harm that is much better found using structured retrospective review of medical records. Healthcare organisations should therefore consider using a portfolio of tools including incident reporting, medical record review and analysis of patient claims to gain a comprehensive picture of safety issues.

Our finding that 20.5% of all discharges experienced at least one AE is in accordance with what has been reported in other studies using GTT.[1] [2] [9–12] Good et al[10] and Landrigan et al[11] reviewed more than 2300 admissions and found that approximately 25% of admissions experienced an AE during hospitalisation. Good et al[10] also found that almost 40% of the AEs were present on admission and that approximately 60% of the AEs occurred during hospitalisation, which is in agreement with our finding that 66% of the AEs occurred and were detected during the hospital stay.

Our finding that patients with an AE had a much longer hospital stay is interesting. In some age groups, the hospital stay for patients with an AE was almost twice as long as in patients without an AE, which is in accordance with the findings of Classen et al.[2] The reason for this result could either be that patients hospitalised for a long period of time are exposed to more risks or that patients who are harmed have a longer hospital stay. We tried to address this question by undertaking a detailed analysis of when the AE occurred. We found that most AEs occurred early in the hospital stay and thus we believe that, in most cases, the AE was the cause and not a consequence of the long hospital stay.

We found that category F harm (ie, patient experienced temporary harm that required initial or prolonged hospitalisation) was most common in contrast to other studies where category E harm was seen most frequently.[1] [10] However, Landrigan et al[11] also found that harm categorised as F, according to the NCC MERP index, was common. The more severe harm categories, G, H and I, added up to 7% in our study, which is equal to or a little lower than previously reported.[1] [11] [12]

More than 40% of the AEs detected were hospital-acquired infections; the most common types were postoperative wound infections and urinary tract infections. Landrigan et al[11] also identified hospital-acquired infections as the most common AEs. Surgical-related and medication-related AEs were also common, which is in accordance with the findings of others.[1] [11] [13] In our study, no significant difference in the AE rate was found between patients older than 65 years and younger patients. This result is in contrast to other larger studies in which elderly patients were found to be at higher risk of AEs.[14] These differences might be explained by the smaller size of our study.

The GTT method, which is recommended by the IHI, does not examine the extent to which an AE is preventable or categorise different types of harm. However, recent studies from the USA include preventability assessments to enhance learning opportunities and guide quality improvement.[9] [10] Landrigan et al[11] reported that internal reviewers rated 63.1% of the AEs found as preventable. Kennerly et al[15] reported that among hospital-acquired AEs, 12.5% were judged to be preventable or probably preventable, and an additional 59% were possibly preventable. In a newly published study, as much as 87% of AEs present on admission to hospital were considered 'preventable/possibly preventable'.[16]

Our assessment that 71% of the AEs identified were preventable is slightly higher than what has been reported previously in two Swedish studies in which 58–70% of the AEs identified were estimated to be preventable.[8] [13] The reason for our finding could be that almost all hospital-acquired infections in our study were classified as preventable.

During the 4-year period, we did not see any reduction in the rate of AEs despite several hospital-wide quality improvement initiatives during this period. Infection control programmes including initiatives directed specifically towards bladder catheterisation and central venous lines together with specially educated physicians on each department for increased focus on hospital-acquired infections and the use of antibiotics. Every ward educated hygiene agents that regularly checked and reported how staff followed rules according to dress code and hand hygiene. Rapid response teams and guidelines for the use of a modified early warning score on every patient were introduced. Another overall hospital initiative was education in communication according to SBAR (Situation, Background, Assessment, Recommendation). A patient safety culture measurement took place in 2010 and included all employees. Based on the results, all departments made an activity plan in order to strengthen the safety culture. These different initiatives were not directly linked to the presented GTT review. However, some departments did their own GTT reviews and their findings resulted in several safety improvement initiatives in their own departments. Our finding of a constant hospital AE rate is in accordance with the results from North Carolina; Landrigan et al[11] studied 10 hospitals over a 6-year period and found little evidence that the rate of harm had decreased substantially over this period. They concluded that penetration of evidence-based safety practices had been quite modest. In contrast, Garrett et al[17] show a progressive reduction in AEs over a 3-year period in a large health system and contribute this achievement to improvement projects based on their findings of major harms. We believe that the implementation rate is slow in our hospital and that the focus should be on translating evidence-based safety interventions into clinical practice.

If we extrapolate our findings of an AE rate of about 20% and 5.4 additional hospital days for patients with an AE to the 32 000 hospital admissions/year, approximately 35 000 additional days are used in the hospital

for treating patients with an AE. Based on our own and other findings, we estimated that about 50–70% of the AEs were preventable, which indicates that annually around 17 500−24 500 additional hospital days are used for caring for patients with a preventable AE. According to the Swedish Association of Local Authorities and Regions, the average cost for a hospital day in Swedish healthcare is SEK8700 (US$1260). Thus, the estimated cost for the hospital could be between SEK152 and SEK213 millions (US$22–31 million) annually. In addition, the cost for ambulatory treatment and further potential hospital visits has to be added. This is a rough estimate of the costs for AEs, but the cost to healthcare is considerable, notwithstanding the effects that AEs have on the patients.

The limitations of this study are that it was conducted in a single hospital and that a relatively small number of medical records were reviewed, which may restrict the generalisability of our findings. The strengths of the study are that the review team was experienced and remained the same throughout the study. In addition, even though the number of medical records reviewed is small, it is a representative sample of the care given at the university hospital.

**Author affiliations**
[1]Division of Health Care Analysis, Department of Medical and Health Sciences, Linköping University, Linköping, Sweden
[2]Public Health Centre, County Council of Östergötland, Linköping, Sweden
[3]Department of Surgery, County Council of Östergötland, Linköping University, Linköping, Sweden
[4]Development and Patient Safety Unit, County Council of Östergötland, Linköping University, Linköping, Sweden
[5]Department of Anesthesia and Intensive Care, County Council of Östergötland, Linköping University, Linköping, Sweden

**Contributors**  HR, RS, PN and LN were responsible for study design, acquisition of the data, analysis, interpretation of the data, drafting the article and intellectual content. MBR and LV were responsible for acquisition of the data, statistical analysis and drafting the article. All authors approved the final version of the manuscript.

**Funding**  This research received no specific grant from any funding agency in the public, commercial or not-for-profit sectors.

**Competing interests**  None.

**Provenance and peer review**  Not commissioned; externally peer reviewed.

**Data sharing statement**  No additional data are available.

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
