## [Reviewer comments · BMJ Open]

Some articles will have been accepted based in part or entirely on reviews undertaken for other BMJ Group journals. These will be reproduced where possible.

ARTICLE DETAILS

TITLE (PROVISIONAL)	Characterisations of adverse events detected in a University Hospital A four year study using the Global Trigger Tool method
AUTHORS	Rutberg, Hans; Borgstedt Risberg, Madeleine; Sjødahl, Rune; Nordqvist, Pernilla; Valter, Lars; Nilsson, Lena

VERSION 1 - REVIEW

REVIEWER	Ellen Tveter Deilkås Centre for Health Services Research, Akershus University Hospital, Norway
REVIEW RETURNED	07-Mar-2014

GENERAL COMMENTS	1. It seems as if the objectives are a bit detached from the main results and discussion of results.a. One objective is to compare voluntarily reported AE's to those identified by medical record review.i. That is a clear and legitimate aim, because this is a different setting from those that such results previously have been reported from.b. To describe the categories of AE's is also a legitimate objective.c. The last objective is less precise: "estimate when the AE occurred in the course of the hospital stay."i. It seems as if this last objective is related to finding out if the prolonged length of stay related to AE's is caused by the AE, or if the AE is the cause of the prolonged length of stay. This should be clearer formulated.2. The first line of the title seems a bit detached from the objectives and results of the article. I suggest making the title more coherent with the article content. This article may be of help: http://tidsskriftet.no/article/29594463. Page 2a. Line 10 : reviews – correct spellingb. Line 17: lacks reference. It does not comply with this article by one of the authors of the GTT white paper (page ii42-ii43) (1).c. Line 51 : review paragraph according to point 2 above.Page 3d. Line 20: a word is missing : departments "were"Page 4e. Line 28 – is unclear, regarding the 2 days. Should also be written as "two" days.Page 8f. Line 52 – improve precision. It is not correct to write "many more". According to page 8 line 38, many AE's are reported voluntarily. This study has shown that many AE's identified by the review are not reported voluntarily. Present wording in line 52 implies that these numbers are compared, which I think would be a misunderstanding.
---

	Page 9 g. Line 26: an explicit explanation of what contribution structured retrospective review of medical records gives to the comprehensive picture of adverse events in healthcare, is necessary to provide this paragraph with necessary logic. h. Last paragraph: Prolonged hospital stay related to AE's is analyzed and interpreted in an interesting way. However the aspect is very different from the methodological aspect commented previously (related to voluntary reporting). The divergence in aspects reduces coherence in the article. It would help if the title, and stating of objectives had bound them together. One should perhaps consider to present and discuss the two aspects in two different articles. Reference List (1) Resar RK, Rozich JD, Classen D. Methodology and rationale for the measurement of harm with trigger tools. Qual Saf Health Care 2003 December 1;12(90002):39ii-45.
--	---

REVIEWER	Persephone Doupi National Institute for Health and Welfare - THL, Helsinki, Finland I have collaborated with one of the authors before (H. Rutberg) and co-authored a paper on use of the GTT-tool in the Nordic countries.
REVIEW RETURNED	29-Apr-2014

GENERAL COMMENTS	Regarding description of the methodology (Checklist point 4): the researchers are comparing the number of AEs identified through a structured review of patient records by the Swedish version of the GTT-tool vs. the number of AEs identified through the voluntary adverse reporting system in use at the University Hospital in Linköping, Sweden. The voluntary reporting system is referred to by its brand name (Synergi®) but there are no details provided about it. It would be good to provide at least some basic information on the system's features and functionality (the minimum would be a link to the company providing it). An important parameter, which is implicitly apparent, is the fact that patient data in the reporting system are identifiable - at least I presume that to be the case, since the researchers were able to check whether an AE identified by the GTT in a patient's record was also included in the reporting system. Generally, that is neither standard functionality of patient safety reporting systems nor standard practice (ie. ability to cross-link information between the reporting and the electronic patient record system databases) and since it considerably affects the type of analysis that can be performed it should be explicitly clarified. Regarding statistical analysis (Checklist point 7): The authors have used the chi-square test to check for possible variations in distribution of AEs across different age groups and by gender. It would have been useful to look also at variations based on specialty. Now information is provided only on the rough distribution between surgical and medical care in the total sample of patient records reviewed, but not with regard to the AEs identified. An additional and rather surprising omission is the investigation of degree of agreement between reviewers both with regard to the
--

	presence of AEs, but even more so on the type of harm and potential preventability of the events. It is well known that the GTT-methodology and overall patient record review as a method for AE identification has been criticized for low or -as a minimum- variable and volatile reviewer reliability. A couple of the pertinent publications are even on the reference list of the paper (#12 and 13) but are not utilized in that perspective. In addition to presenting the percentage of AEs classified as preventable, it would be valuable information for the readers to know the distribution across the different degrees of certainty on preventability that the research team has used. Regarding timeliness and use of references (Checklist point 8): The reference list is good, but there have been several more recent publications on use of the GTT-methodology that the authors could have utilized (incl. publications in BMJ open). Also, as mentioned earlier, some of the papers referred could also be used to provide background for discussion of further aspects, such as inter-rater reliability. The finding that there was no improvement in the rate of AEs over the 4 year period of the study is an important one and deserves further discussion, since it is very meaningful for the sustainability of patient safety work within healthcare provider organisations. The authors point out that similar findings have been reported by other groups, and they underline the need of strengthening the implementation of proven safety interventions into clinical practice. The paper would benefit from some additional information as to how patient safety monitoring activities (such as review with the GTT and use of the voluntary reporting system) are linked with quality improvement initiatives in the specific hospital. The authors mention that during the study period there was an ongoing infection control program, as well as introduction of rapid response teams, however it is not clear whether these were separate or connected activities to patient safety monitoring and if yes, by which means. Overall this is a clear and well-written paper that contributes additional evidence to the ongoing research and implementation work on methods for monitoring and improving patient safety in healthcare organisations.
--	---

VERSION 1 – AUTHOR RESPONSE

Reviewer Ellen Tveter Deilkås

It seems as if the objectives are a bit detached from the main results and discussion of results.

- a. One objective is to compare voluntarily reported AE's to those identified by medical record review.
 - i. That is a clear and legitimate aim, because this is a different setting from those that such results previously have been reported from.
 - b. To describe the categories of AE's is also a legitimate objective.
 - c. The last objective is less precise: "estimate when the AE occurred in the course of the hospital stay."
 - i. It seems as if this last objective is related to finding out if the prolonged length of stay related to AE's is caused by the AE, or if the AE is the cause of the prolonged length of stay. This should be clearer formulated.

Answer: We agree with the referee that the objectives, main results and discussion could be better merged in the manuscript. In order to achieve this we have clarified our aims, both in the abstract and in the introduction. As many others have shown that patients with AEs as a group have longer hospital stays we wanted specifically to address the hypothesis that a prolonged stay is mainly a result of the AE, and not the other way around - a longer stay exposes a patient to an increased risk of AEs. We have tried to express this more clearly in the revised version.

The first line of the title seems a bit detached from the objectives and results of the article. I suggest making the title more coherent with the article content. This article may be of help:

<http://tidsskriftet.no/article/2959446>

Answer: We have changed the title to be more informative about the whole study, and not describe a specific result.

Page 2 a. Line 10 : reviews – correct spelling

Answer: The spelling has been corrected

Line 17: lacks reference. It does not comply with this article by one of the authors of the GTT white paper (page ii42-ii43) (1).

(1) Resar RK, Rozich JD, Classen D. Methodology and rationale for the measurement of harm with trigger tools. Qual Saf Health Care 2003 December 1;12(90002):39ii-45.

Answer: We think that the referee have the expression “The GTT was primarily designed as..” in mind. We agree and have change the sentence to “The GTT can be used as...” and we refer to the GTT White paper from 2009 (REF #6).

Line 51 : review paragraph according to point 2 above.

Answer: Please, se our answer to the comment regarding aim, results and discussion.

Page 3 d. Line 20: a word is missing : departments “were”

Answer: We disagree, but have changed the sentence to make it easier to read.

Page 4 e. Line 28 – is unclear, regarding the 2 days. Should also be written as “two” days.

Answer: We have rewritten in order to clarify.

Page 8 f. Line 52 – improve precision. It is not correct to write “many more”. According to page 8 line 38, many AE’s are reported voluntarily. This study has shown that many AE’s identified by the review are not reported voluntarily. Present wording in line 52 implies that these numbers are compared, which I think would be a misunderstanding.

Answer: We agree with the referee, and have changed the sentence to prevent misunderstanding.

Page 9 g. Line 26: an explicit explanation of what contribution structured retrospective review of medical records gives to the comprehensive picture of adverse events in healthcare, is necessary to provide this paragraph with necessary logic.

Answer: We have added the description that record review contributes with a better detection of harm and therefore contributes to a comprehensive picture of safety issues in healthcare.

Last paragraph: Prolonged hospital stay related to AE’s is analyzed and interpreted in an interesting way. However the aspect is very different from the methodological aspect commented previously (related to voluntary reporting). The divergence in aspects reduces coherence in the article. It would help if the title, and stating of objectives had bound them together. One should perhaps consider to present and discuss the two aspects in two different articles.

Answer: We have, as described in earlier responds, changed the title and clarified our objectives.

With these changes, we don´t think that the aspects of prolonged hospital stay and the connected

discussion is superfluous and that the two aspects should be divided into separate articles.

Reviewer Persephone Doupi

Regarding description of the methodology (Checklist point 4):

the researchers are comparing the number of AEs identified through a structured review of patient records by the Swedish version of the GTT-tool vs. the number of AEs identified through the voluntary adverse reporting system in use at the University Hospital in Linköping, Sweden. The voluntary reporting system is referred to by its brand name (Synergi®) but there are no details provided about it. It would be good to provide at least some basic information on the system's features and functionality (the minimum would be a link to the company providing it).

Answer: We have provided additional information about the Synergi system.

An important parameter, which is implicitly apparent, is the fact that patient data in the reporting system are identifiable - at least I presume that to be the case, since the researchers were able to check whether an AE identified by the GTT in a patient's record was also included in the reporting system. Generally, that is neither standard functionality of patient safety reporting systems nor standard practice (ie. ability to cross-link information between the reporting and the electronic patient record system databases) and since it considerably affects the type of analysis that can be performed it should be explicitly clarified.

Answer: Patient data are often included in the reporting system, and always when hospital-acquired infections are reported. The most effective way to look for the AEs in the voluntary reporting system was to search for AEs reported by the department at the time the patient stayed at the hospital and check for consistency. From the medical record we knew the type and date for the AE. In the reporting system all reported AEs are shortly described and given a heading. As reporting is sometimes delayed, we had to look in a broader time-frame than only the days of hospitalisation. This is described in the revised version.

Regarding statistical analysis (Checklist point 7): The authors have used the chi-square test to check for possible variations in distribution of AEs across different age groups and by gender. It would have been useful to look also at variations based on specialty. Now information is provided only on the rough distribution between surgical and medical care in the total sample of patient records reviewed, but not with regard to the AEs identified.

Answer: We have divided the types of AEs in two groups – surgical care and medical care and show the separated results in a new Fig 2. There were significantly more AEs in connection with surgical care, and we added this information in the result part. However, if we further divide the AEs into subgroups based on medical specialty (orthopedics, cardiology etc) the groups will include so small numbers of AEs that a statistical calculation is not possible.

An additional and rather surprising omission is the investigation of degree of agreement between reviewers both with regard to the presence of AEs, but even more so on the type of harm and potential preventability of the events. It is well known that the GTT-methodology and overall patient record review as a method for AE identification has been criticized for low or -as a minimum- variable and volatile reviewer reliability. A couple of the pertinent publications are even on the reference list of the paper (#12 and 13) but are not utilized in that perspective.

Answer: We agree with the referee that reviewer variable reliability is a criticised factor in the GTT methodology. In this article our aim was to describe the findings from a continuous GTT review in a university hospital over a period of four years. We describe our process under methods and that did not include a comparison between the primary nurse reviewers. We also describe that the review team was experienced and stable over the study period. The team had regular discussions and came to consensus regarding the AEs. Two recent studies (added as REF #16 and #17) characterising AEs in large health care systems and using professional nurse reviewers tested inter-rater reliability on a

small part of their records. Their main purpose was to give a view over the characteristics of AE over a three - five-year period, not to evaluate the methodology. Our aim was similar. Although, we agree that not have included degree of agreement is a limitation, we don't believe that it has an impact on our main results.

In addition to presenting the percentage of AEs classified as preventable, it would be valuable information for the readers to know the distribution across the different degrees of certainty on preventability that the research team has used.

Answer: We agree with the referee and have added these results.

Regarding timeliness and use of references (Checklist point 8): The reference list is good, but there have been several more recent publications on use of the GTT-methodology that the authors could have utilized (incl. publications in BMJ open). Also, as mentioned earlier, some of the papers referred could also be used to provide background for discussion of further aspects, such as inter-rater reliability.

Answer: We have added two new references, REF #16 and #17 in the revised version and expanded somewhat in the discussion section. As explained above we have not discussed inter-rater reliability.

The finding that there was no improvement in the rate of AEs over the 4 year period of the study is an important one and deserves further discussion, since it is very meaningful for the sustainability of patient safety work within healthcare provider organisations. The authors point out that similar findings have been reported by other groups, and they underline the need of strengthening the implementation of proven safety interventions into clinical practice. The paper would benefit from some additional information as to how patient safety monitoring activities (such as review with the GTT and use of the voluntary reporting system) are linked with quality improvement initiatives in the specific hospital. The authors mention that during the study period there was an ongoing infection control program, as well as introduction of rapid response teams, however it is not clear whether these were separate or connected activities to patient safety monitoring and if yes, by which means.

Answer: We have given more information of quality improvement activities in our hospital that were undertaken during the period for our study. These activities were not directly linked to the GTT review.

Overall this is a clear and well-written paper that contributes additional evidence to the ongoing research and implementation work on methods for monitoring and improving patient safety in healthcare organisations.